

# Proteomic analysis reveals some common proteins in the kidney stone matrix

Yuanyuan Yang[1], Senyuan Hong[1], Cong Li[1], Jiaqiao Zhang[1], Henglong Hu[1], Xiaolong Chen[2], Kehua Jiang[2], Fa Sun[2], Qing Wang[2,3] and Shaogang Wang[1]

[1] Department of Urology, Tongji Hospital, Tongji Medical College, Huazhong University of Science and Technology, Wuhan, Hubei, China
[2] Department of Urology, Guizhou Provincial People's Hospital, Guizhou University, Guiyang, Guizhou, China
[3] Department of Research Laboratory Center, Guizhou Provincial People's Hospital, Guizhou University, Guiyang, Guizhou, China

## ABSTRACT

**Background:** Proteins are the most abundant component of kidney stone matrices and their presence may reflect the process of the stone's formation. Many studies have explored the proteomics of urinary stones and crystals. We sought to comprehensively identify the proteins found in kidney stones and to identify new, reliable biomolecules for use in nephrolithiasis research.

**Methods:** We conducted bioinformatics research in November 2020 on the proteomics of urinary stones and crystals. We used the ClusterProfiler R package to transform proteins into their corresponding genes and Ensembl IDs. In each study we located where proteomic results intersected to determine the 20 most frequently identified stone matrix proteins. We used the Human Protein Atlas to obtain the biological information of the 20 proteins and conducted Gene Ontology (GO) and Kyoto Encyclopedia of Genes and Genome (KEGG) analysis to explore their biological functions. We also performed immunohistochemistry to detect the expression of the top five stone matrix proteins in renal tissue.

**Results:** We included 19 relevant studies for analysis. We then identified 1,409 proteins in the stone matrix after the duplicates were removed. The 20 most-commonly identified stone matrix proteins were: S100A8, S100A9, uromodulin, albumin, osteopontin, lactotransferrin, vitamin K-dependent protein Z, prothrombin, hemoglobin subunit beta, myeloperoxidase, mannan-binding lectin serine protease 2, lysozyme C, complement C3, serum amyloid *P*-component, cathepsin G, vitronectin, apolipoprotein A-1, eosinophil cationic protein, fibrinogen alpha chain, and apolipoprotein D. GO and KEGG analysis revealed that these proteins were typically engaged in inflammation and immune response. Immunohistochemistry of the top five stone matrix proteins in renal tissue showed that the expression of S100A8, S100A9, and osteopontin increased, while uromodulin decreased in kidney stone patients. Albumin was rarely expressed in the kidney with no significant difference between healthy controls and kidney stone patients.

**Conclusion:** Proteomic analysis revealed some common inflammation-related proteins in the kidney stone matrix. The role of these proteins in stone formation should be explored for their potential use as diagnostic biomarkers and therapeutic targets for urolithiasis.

Corresponding authors
Qing Wang,
wangqingtjm@hust.edu.cn
Shaogang Wang,
sgwangtjm@163.com

## INTRODUCTION

Kidney stones are a common public health problem. It is estimated that 10% to 12% of the general population will suffer from nephrolithiasis in their lifetime (*Coe, Evan & Worcester, 2005*). The prevalence of kidney stones is currently 6.4% of the general population in China and this incidence has increased over the past four decades in Western countries (*Zeng et al., 2017*; *Scales et al., 2012*). The 10-year recurrence rate for kidney stones may be as high as 50%, resulting in repetitive treatments and a huge economic burden (*Uribarri, Oh & Carroll, 1989*; *Geraghty et al., 2020*). Thus, it is important to explore the pathogenesis of nephrolithiasis and provide a theoretical basis for its treatment and prevention.

Kidney stones are composed of 97–98% mineral salts and 2–3% organic matrix (*Boyce, 1968*). The organic matrix, including proteins, lipids, glycosaminoglycans, and carbohydrates, plays a role in modulating the formation of stones (*Boyce & Garvey, 1956*). Proteins are the most abundant component in a kidney stone and comprise approximately 64% of the stone matrix (*Boyce, 1968*). The identification of the matrix proteins contributes to a better understanding of a stone's formation. Advancements in proteomic techniques and the introduction of mass spectrometry have improved the study of the proteomics in urinary stones and crystals (*Mushtaq et al., 2007*; *Canales et al., 2008*; *Merchant et al., 2008*; *Chen et al., 2008*; *Canales et al., 2009*; *Thurgood & Ryall, 2010*; *Thurgood et al., 2010*; *Canales et al., 2010*; *Kaneko et al., 2011*; *Jou et al., 2012*; *Kaneko et al., 2012*; *Okumura et al., 2013*; *Boonla et al., 2014*; *Kaneko et al., 2014*; *Kaneko et al., 2015*; *Martelli et al., 2016*; *Witzmann et al., 2016*; *Kaneko et al., 2018*; *Wesson et al., 2019*). More than one thousand proteins have been detected in the stone matrix to date. Some common proteins, including S100A8, S100A9, and osteopontin (OPN) are frequently detected in stones and may reveal a potentially universal pattern for stone formation.

We conducted a systemic review of studies that focused on the proteomics of the stone matrix to determine the proteins in kidney stones and to identify those that were the most frequently occuring. We performed bioinformatic analysis to explore the function of the top 20 matrix proteins and immunohistochemistry to detect the expression of the top five stone matrix proteins in renal tissue. We sought to provide new and reliable biomolecules for urolithiasis research.

## METHODS

### Literature search

We conducted a systematic literature search of Medline, Embase, and the Web of Science databases in November 2020. The following search strategy was used: (((((urin*) OR kidney) OR renal)) AND (((((stone) OR calculi) OR calcium) OR matrix) OR crystal)) AND proteomic. We included studies that were written in English and associated with proteomics of urinary stones and crystals in our study.

## Bioinformatic analysis

We used clusterProfiler package in R version 4.0.0 to transform the proteins to related genes and Ensembl IDs (*Yu et al., 2012*) to unify the names of proteins in different studies. We took the intersection of the proteomic results from each study and identified the 20 most common stone matrix proteins. We searched the Human Protein Altas to obtain the biological information of the matrix proteins (https://www.proteinatlas.org/) (*Uhlén et al., 2015*). Gene Ontology (GO) and Kyoto Encyclopedia of Genes and Genome (KEGG) analyses were performed using the OmicShare tool, a free online platform for data analysis (http://www.omicshare.com/tools) to explore the biological function of the 20 proteins. Enrichment results were filtered with a false discovery rate (FDR) of < 0.05.

## Immunohistochemistry

Renal tissue was collected from patients undergoing nephrectomy due to kidney stones or renal tumor. Tissue from kidney stone patients was classified as the stone group and normal paracancer tissue from kidney tumor patients was defined as the control group. The samples were fixed with formalin and embedded in paraffin for routine sectioning. Slices of the tissue were incubated with anti-S100A8 antibody (ab92331; Abcam, Cambridge, UK), anti-S100A9 antibody (ab92507; Abcam, Cambridge, UK), anti-uromodulin antibody (A01303-2; Boster, Selangor, Malaysia), anti-OPN (ab8448; Abcam, Cambridge, UK), and anti-albumin antibody (A1363; Abclonal, Wuhan, China) for immunohistochemistry. We used Image J software version 1.52 (National Institute of Mental Health, Bethesda, MD, USA) was used to quantify the relative area of the positive staining area. The ethical review board of Tongji Hospital, Tongji Medical College, Huazhong University of Science and Technology approved the collection and use of tissue samples (2019S1147). The written form of informed consent was obtained from all patients.

## Statistical analysis

Measurement data are presented as mean ± standard deviations. A Student's t-test was conducted using Prism 9.0 for statistical analysis. A *p*-value 0.05 was considered to be statistically significant.

# RESULTS

## An overview of the included studies

Figure 1 shows the process used to select studies. We included 19 studies exploring the proteomics of urinary stones and crystals and an overview of these studies is shown in Table 1. Liquid chromatography-tandem mass spectrometry (LC-MS/MS) was the most commonly-used proteomic technique and provided high-throughput amino acid sequence data. Most studies focused on calcium oxalate (CaOx) and uric acid (UA) stones but occasionally rare stone types such as matrix and CaCO3 stones were also studied. Thurgood et al. also reported the protein files in urinary crystals from a healthy population (*Thurgood & Ryall, 2010*; *Thurgood et al., 2010*). Many common inflammatory proteins

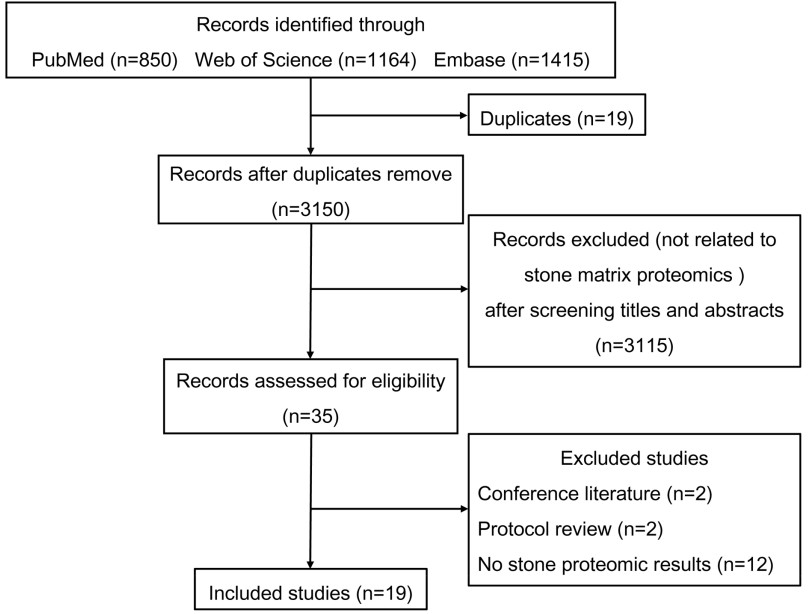

**Figure 1 Flowchart of the literature search and study selection.**

were identified in various studies, indicating the involvement of inflammation in stone formation.

## Identifying the 20 most common proteins in stone matrices

The detailed proteomic results of each study are presented in File 1. A total of 1,409 proteins were detected in the stone matrix after removing duplicates (File 1). We intersected each study to identify the most common stone matrix proteins. The most frequently detected proteins in the stone matrix were: S100A8, S100A9, uromodulin, albumin, OPN, lactotransferrin, vitamin K-dependent protein Z, prothrombin, hemoglobin subunit beta, myeloperoxidase, mannan-binding lectin serine protease 2, lysozyme C, complement C3, serum amyloid *P*-component, cathepsin G, vitronectin, apolipoprotein A-1, eosinophil cationic protein, fibrinogen alpha chain, and apolipoprotein D. Figure 2 shows the exact detection frequency of each protein. We also conducted a subgroup analysis of the studies and focused only on CaOx stones. We found that the 20 most common proteins in CaOx stone matrices were same as the above 20 proteins (File 2). We searched the online Human Protein Altas database to explore the biology of these proteins. Uromodulin, OPN, lysozyme C, and apolipoprotein D had medium to high expression in renal tubular epithelial cells, while the remaining proteins were rich in other tissues (Table 2).

## Biological function of the 20 most common proteins in stone matrices

GO annotation showed that the top 20 proteins were involved in the stimulus response, binding activity, and extracellular region. They played a role in the immune response, indicating that stone formation was associated with the inflammatory response.

**Table 1 Basic information of included studies.** Detailed information of the 19 studies we included in our research.

| Years | Authors | Sample size | Stone composition | Proteomic technique | Verification methods | Proteins identified | Main findings |
|---|---|---|---|---|---|---|---|
| 2007 (*Mushtaq et al., 2007*) | Mushtaq et al. | 40 | CaOx | 1D PAGE LC-MS/MS | Western blot | 4 | Myeloperoxidase, α-defensin and calgranulin were identified from inner core of CaOx stones and they promoted the aggregation of CaOx crystals. Osteopontin was detected both in the inner and outer matrix of CaOx stones. |
| 2008 (*Canales et al., 2008*) | Canales et al. | 7 | CaOx | LC-MS/MS | NA | 68 | A significant number of inflammatory proteins, such as immunoglobulin, α-defensin-3, clusterin, complement C3a, kininogen, calgranulin and fibrinogen, were found in CaOx stones matrix. |
| 2008 (*Merchant et al., 2008*) | Merchant et al. | 4 | CaOx | LC-MS/MS | Western blot | 158 | A total of 58 prevalent proteins were detected in at least two of the three LC-MS/MS analyses. Pathway analysis suggested that a significant fraction of CaOx stone matrix proteins participate in inflammatory processes. |
| 2008 (*Chen et al., 2008*) | Chen et al. | 10 | CaOx | 1D PAGE LC-MS/MS | NA | 11 | There were abundant proteins with molecular weight around 27, 14, and 10 kDa in CaOx stones matrix. Methylation, deamidation, and oxidation were indentified with mass spectroscopy in these proteins. |
| 2009 (*Canales et al., 2009*) | Canales et al. | 1 | Matrix stone | LC-MS/MS | NA | 33 | Protein file of matrix stones included many similar inflammatory proteins seen in previous proteomic studies of CaOx stone matrix, indicating a primary inflammatory mechanism behind matrix stones. |
| 2010 (*Thurgood & Ryall, 2010*) | Thurgood et al.[#] | 5 | HA | LC-MS/MS | NA | 36 | Binding of proteins to urinary hydroxyapatite, brushite, and uric acid crystals is selective and distinct. Several proteins consistently detected in the healthy urine crystal extracts, such as osteopontin, prothrombin and S100A9, have been previously implicated in kidney stone disease. |
|  |  |  | Brushite | LC-MS/MS | NA | 65 |  |
|  |  |  | UA | LC-MS/MS | NA | 7 |  |
| 2010 (*Thurgood et al., 2010*) | Thurgood et al.[#] | 5 | COM | LC-MS/MS | 2D SDS-PAGE | 14 | The incorporation of proteins into COM and COD crystals from healthy human urine was selective. Principal proteins in COM crystal extracts were prothrombin fragment 1, S100A9, and IGκV1-5, while those in COD crystals included osteopontin, IGκV1-5, S100A9, annexin A1, HMW kininogen-1, and inter-α-inhibitor. |
|  |  |  | COD | LC-MS/MS | NA | 34 |  |
| 2010 (*Canales et al., 2010*) | Canales et al. | 13 | CaOx | LC-MS/MS | NA | 49 | CaOx and CaP stones shared similar matrix proteins associated with inflammatory response, indicating that inflammation play an important role in calcium stone formation, no matter as an origin role or a secondary response. |
|  |  | 12 | CaP | LC-MS/MS | NA | 45 |  |
| 2011 (*Kaneko et al., 2011*) | Kaneko et al. | 1 | UA, COM | LC-MS/MS | NA | 32 | Calcium-binding proteins, such as calprotectin, psoriasin, calprotectin and so on, were identified in stones from patients with hyperuricemia. They may play a significant role in the formation of kidney uric acid stones. |
| 2012 (*Jou et al., 2012*) | Jou et al. | 5 | UA | LC-MS/MS | Western blot | 242 | The function of proteins identified from uric acid stones is mainly engaged in inflammatory process and lipid metabolism, implying a possible relation between lipotoxicity and stone formation. |

(Continued)

| Years | Authors | Sample size | Stone composition | Proteomic technique | Verification methods | Proteins identified | Main findings |
|---|---|---|---|---|---|---|---|
| 2012 (*Kaneko et al., 2012*) | Kaneko et al. | 17 | CaOx, UA | 1D PAGE LC-MS/MS | Western blot | 30 | Uromodulin and albumin are often detected in stones. Osteopontin, prothrombin, protein S and protein Z are identified specifically in calcium oxalate stones. Immunoglobin G fragments are detected in uric acid stones. |
| 2013 (*Okumura et al., 2013*) | Okumura et al. | 9 | CaOx | LC-MS/MS | Western blot | 92 | Prothrombin, osteopontin, S100A8 and S100A9 were found in most stones, some samples had high contents of prothrombin and osteopontin, while others had high contents of calgranulins and neutrophil-enriched proteins. |
| 2014 (*Boonla et al., 2014*) | Boonla et al. | 16 | COM, UA, MAP | 1D PAGE LC-MS/MS | Western blot | 62 | Kidney stones greatly contained inflammatory and fibrotic proteins, indicating that inflammation and fibrosis are involved in the formation of stones. S100A8 and fibronectin were the most abundant protein in stone matrix. |
| 2014 (*Kaneko et al., 2014*) | Kaneko et al. | 1 | CaCO3, CaOx | 1D PAGE LC-MS/MS | NA | 53 | Matrix proteins from calcium carbonate stone are mostly associated with cell adhesion and cytoskeleton. These identified proteins may play an important role on urolithiasis in alkaline condition. |
| 2015 (*Kaneko et al., 2015*) | Kaneko et al. | 16 | COM, COD, HA | 1D PAGE LC-MS/MS | NA | 65 | Many plasma proteins were frequently detected in stone matrix regardless of the stone components. Identified proteins were involved in inflammation, coagulation process, and osteometabolism. |
| 2016 (*Martelli et al., 2016*) | Martelli et al. | 4 | Matrix stone | 1D PAGE LC-MS/MS | NA | 142 | S100A8, S100A9 and neutrophil defensin were identified as the main component of matrix stones. Inflammatory process may be the origin of this kind of rare soft calculi formation but not be the consequence. |
| 2016 Witzmann et al. | 2 | CaOx | LC-MS/MS | NA | 1059 | A more | (*Witzmann et al., 2016*) complex stone matrix proteome than previously studies was reported. Matrix proteins were related to immune response, inflammation, injury, and tissue repair. |
| 2018 (*Kaneko et al., 2018*) | Kaneko et al. | 1 | COM, UA | 1D PAGE LC-MS/MS | NA | 59 | Proteins relevant to cell adhesion, self-defense, and plasma commonly play a major role in the generation of stone. The proteins in the interface likely function to enlarge the stone *via* the addition of different crystals. |
| 2019 (*Wesson et al., 2019*) | Wesson et al. | 8 | CaOx | LC-MS/MS | NA | 366 | Osteopontin, mannan-binding lectin serine protease 2, vitamin K-dependent protein Z, prothrombin, and hemoglobin β chain were prominently enriched in matrix, accounting for a mass fraction of >30% of matrix protein. Many identified matrix proteins are reported in intracellular or nuclear locations, indicating a significant role of cell injury in stone formation. |

**Notes:**
[#] Crystals are isolated from the urine of healthy people without urinary stones.
COM, Calcium oxalate monohydrate; COD, Calcium oxalate dihydrate; CaCO3, Calcium carbonate; CaOx, Calcium oxalate; CaP, Calcium phosphate; UA, Uric acid; MAP, Magnesium ammonium phosphate; HA, Hydroxyapatite.

These 20 proteins showed a strong ability to bind with calcium ions, which may explain why they appeared in the stone matrix (Fig. 3). KEGG annotation also showed that the top 20 proteins participated in immune and infectious disease responses. They were

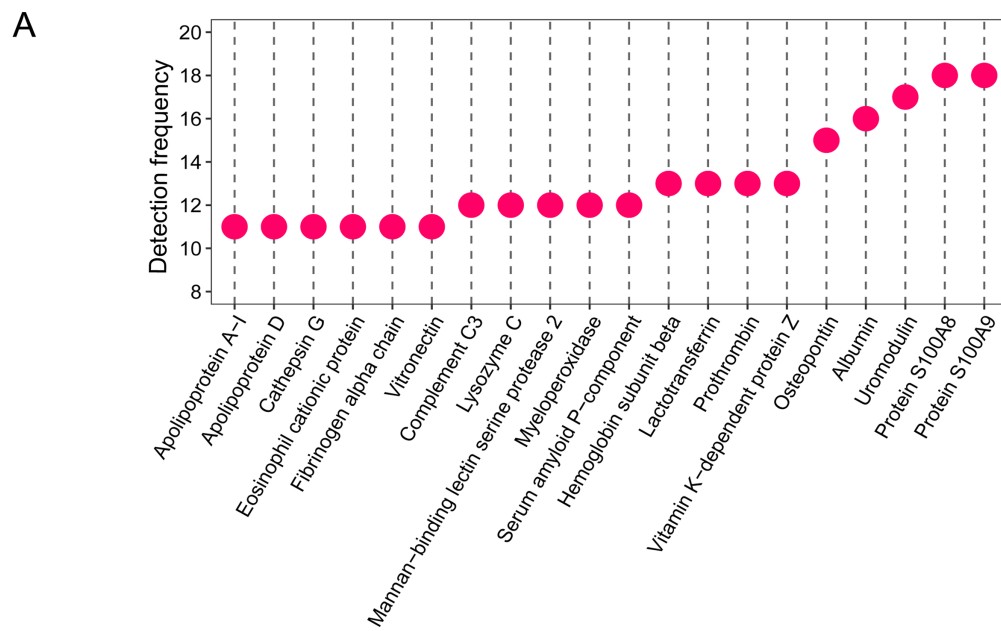

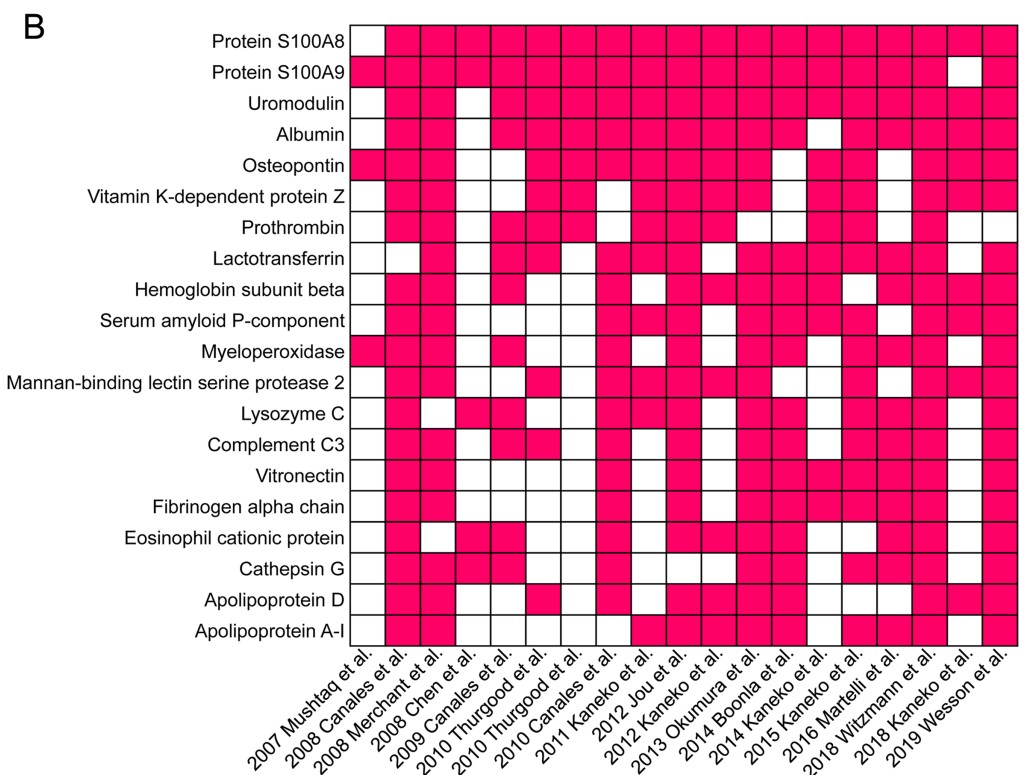

**Figure 2 The 20 most common proteins in stone matrix.** (A) Detected frequency of the top 20 proteins in stone matrix. (B) Detection of the top 20 proteins in each study.

**Table 2 Biological information of the 20 most common proteins in stone matrix.** Biological information of the 20 most common proteins in stone matrix (from the Human Protein Altas).

| Proteins | GENE | Tissue specificity | Blood specificity | Expression in glomeruli | Expression in renal tubules | Biological process |
|---|---|---|---|---|---|---|
| S100A8 | S100A8 | Blood, bone marrow, esophagus, tongue | Neutrophil, classical monocyte | Rare | Rare | Apoptosis, autophagy, chemotaxis, immunity, inflammatory response, innate immunity |
| S100A9 | S100A9 | Blood, bone marrow, esophagus, tongue | Neutrophil, classical monocyte | Rare | Rare | Apoptosis, autophagy, chemotaxis, immunity, inflammatory response, innate immunity |
| Uromodulin | UMOD | Kidney | None | Rare | High | Ciliopathy, disease mutation, nephronophthisis |
| Albumin | ALB | Liver | Naive CD4 T-cell | Rare | Rare | Cancer-related genes, disease mutation |
| Osteopontin | SPP1 | Gallbladder, kidney, placenta | Neutrophil | Rare | High | Biomineralization, cell adhesion |
| Lactotransferrin | LTF | Bone marrow, salivary gland, seminal vesicle | Non-classical monocyte, neutrophil | Rare | Rare | Immunity, ion transport, osteogenesis, iron transport, transcription, transcription regulation, transport |
| Vitamin K-dependent protein Z | PROZ | Liver | Intermediate monocyte, T-reg | Rare | Rare | Blood coagulation, hemostasis |
| Prothrombin | F2 | Liver | None | Rare | Rare | Acute phase, blood coagulation, hemostasis |
| Hemoglobin subunit beta | HBB | Bone marrow | Neutrophil, plasmacytoid DC | Rare | Rare | Oxygen transport, transport |
| Myeloperoxidase | MPO | Bone marrow | Classical monocyte, intermediate monocyte, myeloid DC, neutrophil | Rare | Rare | Hydrogen peroxide |
| Mannan-binding lectin serine protease 2 | MASP2 | Liver | Low immune cell specificity | Rare | Rare | Complement pathway, immunity, innate immunity |
| Lysozyme C | LYZ | Blood, salivary gland | Classical monocyte, myeloid DC | Rare | Medium | Amyloidosis, disease mutation |
| Complement C3 | C3 | Liver | Non-classical monocyte | Rare | Rare | Complement alternate pathway, complement pathway, fatty acid metabolism, host-virus interaction, immunity, inflammatory response, Innate immunity, lipid metabolism |
| Serum amyloid P-component | APCS | Liver | None | Rare | Rare | Calcium, lectin, metal-binding |
| Cathepsin G | CTSG | Bone marrow | Neutrophil, classical monocyte, plasmacytoid DC, NK-cell, myeloid DC, memory CD8 T-cell, naive CD8 T-cell | Rare | Rare | Antibiotic, antimicrobial, hydrolase, protease, serine protease |

| Proteins | GENE | Tissue specificity | Blood specificity | Expression in glomeruli | Expression in renal tubules | Biological process |
|---|---|---|---|---|---|---|
| Vitronectin | VTN | Liver | Naive CD8 T-cell | Rare | Rare | Cell adhesion |
| Apolipoprotein A-1 | APOA1 | Liver | Plasmacytoid DC | Rare | Rare | Cholesterol metabolism, lipid metabolism, lipid transport, steroid metabolism, transport, sterol metabolism |
| Eosinophil cationic protein | RNASE3 | Blood, bone marrow | Eosinophil | Rare | Rare | Antibiotic, antimicrobial, endonuclease, hydrolase, nuclease |
| Fibrinogen alpha chain | FGA | Liver | None | Rare | Rare | Adaptive immunity, blood coagulation, hemostasis, immunity, innate immunity |
| Apolipoprotein D | APOD | Breast | Memory B-cell | Rare | High | Transport |

enriched in the complement and coagulation cascades pathway, which activates the complements involved in the immune response and coagulation cascades associated with inflammation (Fig. 4).

## The expression of the top five stone matrix proteins in renal tissue

Among the 20 common stone matrix proteins, S100A8 and S100A9 were the most frequently detected and they appeared in all kinds of urinary stones. Our previous study demonstrated that urinary exosomes from kidney stone patients were rich in S100A8 and S100A9 (*Wang et al., 2020*). Immunohistochemistry results showed that the expression of S100A8, S100A9 and OPN was significantly increased in renal tissue from kidney stone patients. In contrast, the expression of uromodulin was decreased in the renal tissue of kidney stone patients. Positive S100A8 and S100A9 staining were restricted in cells within vessels in normal kidney tissues. However, S100A8 and S100A9 were mainly expressed in the renal interstitium of kidney stone patients. The expression of OPN and uromodulin typically occured in renal tubular epithelial cells in both the control and stone groups. Albumin was rarely detected in the kidney and there was no significant difference between healthy controls and kidney stone patients (Fig. 5).

## DISCUSSION

Stone matrix proteins play a role in modulating the nucleation, aggregation and growth of urinary crystals, which may affect stone formation (*Narula et al., 2020*). Several investigators in the last 20[th] century attempted to identify the proteins in stones, but were limited by the lack of high throughput technology (*Binette et al., 1996*; *Sugimoto et al., 1985*; *Jones & Resnick, 1990*). Thousands of proteins have been identified in the stone matrix with the introduction of mass spectrometry, which has improved the understanding of the pathogenesis of urolithiasis.

The binding of proteins to various stones is selective despite the presence of some common proteins. We identified the 20 most common proteins in the stone matrix. These

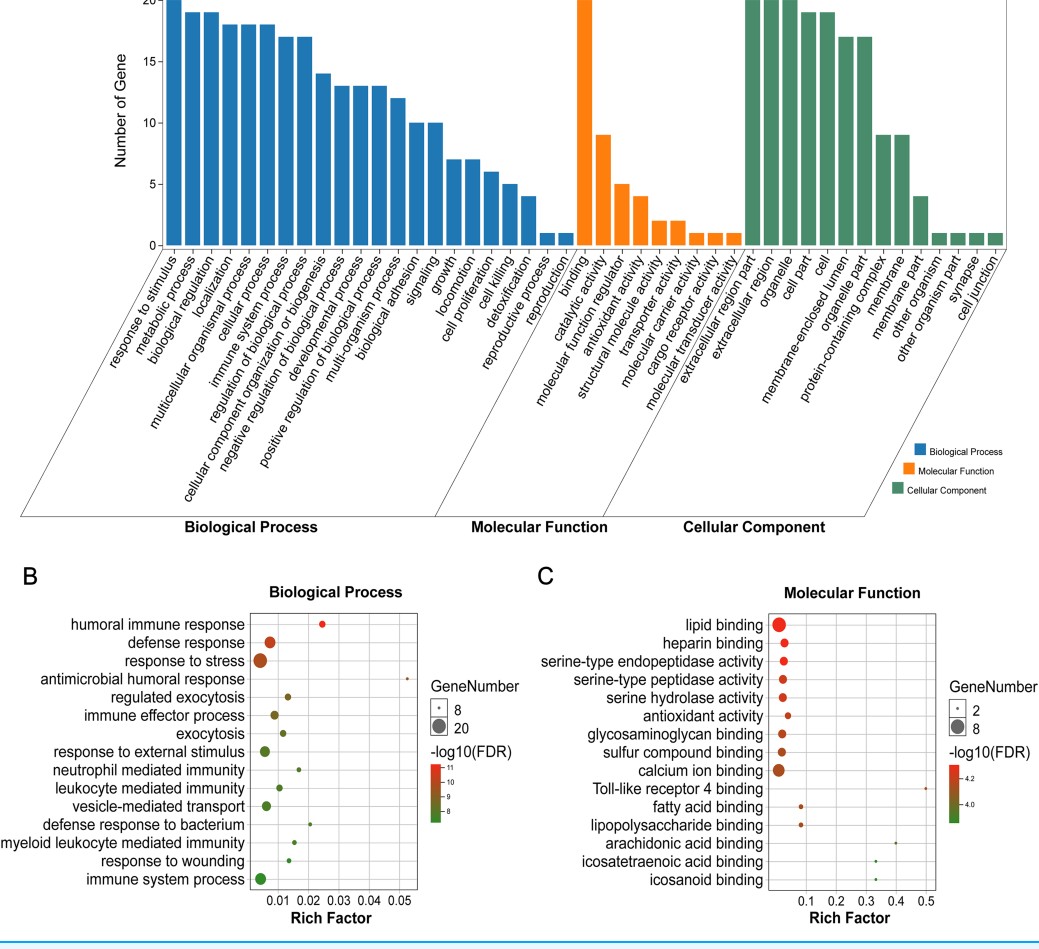

**Figure 3 GO analysis of the 20 most common proteins in stone matrix.** (A) GO annotation. (B) GO biological process enrichment analysis. (C) GO molecular function enrichment analysis. Rich Factor referred to the ratio of the number of enriched genes in the GO category to the total genes in that category. FDR referred to false discovery rate. FDR < 0.05 was set as the cut-off value.

proteins typically engaged in immune and inflammatory responses, indicating the role of inflammation in stone formation. Recent studies have reported that the macrophage-related immune response participates in stone formation and immunotherapy may be used to treat kidney stone disease (*Dominguez-Gutierrez et al., 2020*). Only four of the 20 most common matrix proteins showed medium-to-high expression in the kidney (Table 2). Other non-renal-specific proteins were thought to originate from plasma proteins physiologically filtered through the glomeruli, products of immune cells infiltrated in the renal interstitium, or plasma proteins pathologically exudated due to the injury and infection caused by stones.

S100A8 and S100A9 were detected with the highest frequency from the top five matrix proteins. These proteins belong to the S100 calcium binding family and are primarily derived from neutrophils, monocytes and M1 macrophages (*Pruenster et al., 2016*; *Dessing et al., 2015*). As critical alarmin, S100A8 and S100A9 play an important role in regulating

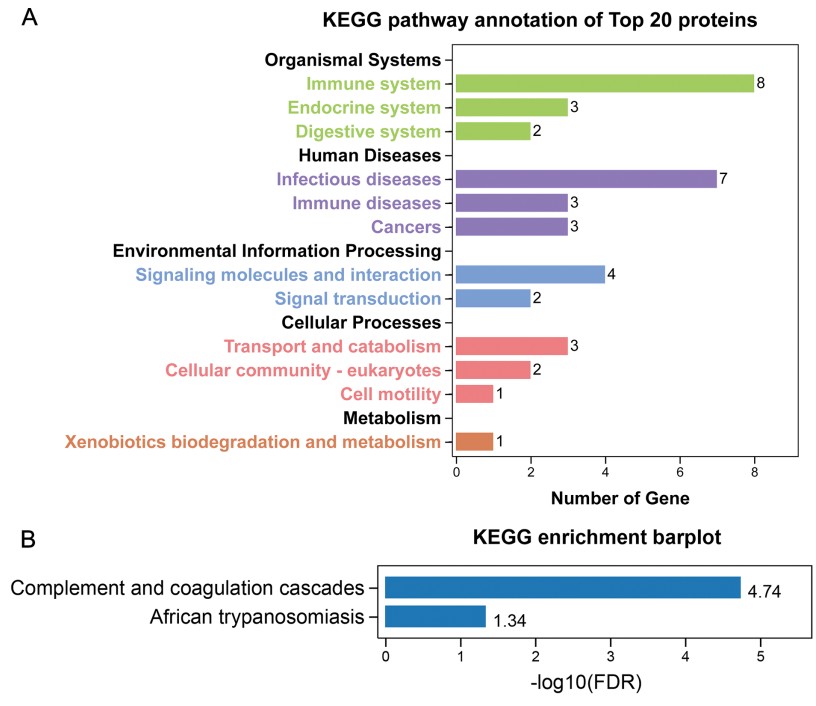

**Figure 4 KEGG analysis of the 20 most common proteins in stone matrix.** (A) KEGG annotation. (B) KEGG disease enrichment analysis.

the immune response. They mediate the production of proinflammatory cytokines and the recruitment of leukocytes (*Wang et al., 2018*). *New et al. (2013)* reported that S100A9-rich vesicles from macrophages have a powerful potential for calcification in 2013. We previously demonstrated that the expression of urinary exosomal S100A8 and S100A9 in kidney stone patients was higher than in healthy controls (*Wang et al., 2020*). In this study, the expression of S100A8 and S100A9 in renal tissue was also found to be elevated in kidney stone patients. In addition, S100A8 and S100A9 were detected in the renal interstitium of kidney stone patients, which may originate from macrophages in the kidney. The potential role of S100A8 and S100A9 in stone formation is worthy of further exploration. They may be valuable as both diagnostic biomarkers and therapeutic targets for urolithiasis.

Uromodulin, also known as Tamm-Horsfall protein (THP), is the third most common stone matrix protein. It is secreted from the thick ascending limb of Henle's loop (*Gokhale, Glenton & Khan, 2001*). Previous *in vitro* studies have shown that THP could inhibit the aggregation of calcium oxalate and calcium phosphate crystals (*Kumar & Lieske, 2006*). *Liu et al. (2010)* identified crystals deposited in the kidneys of THP knockout mice at as early as two months of age. Transmission electron microscopy showed that the deposits were spherical in shape with multiple layers, which is similar to human calcium oxalate stones (*Liu et al., 2010*). THP is thought to play an inhibitory role in stone formation. However, urinary THP excretion decreases in kidney stone patients (*Lau et al., 2008*). We also found that the expression of uromodulin decreased in the renal tissue of kidney stone patients. Stone formers have been reported to excrete defective THP, which lacks

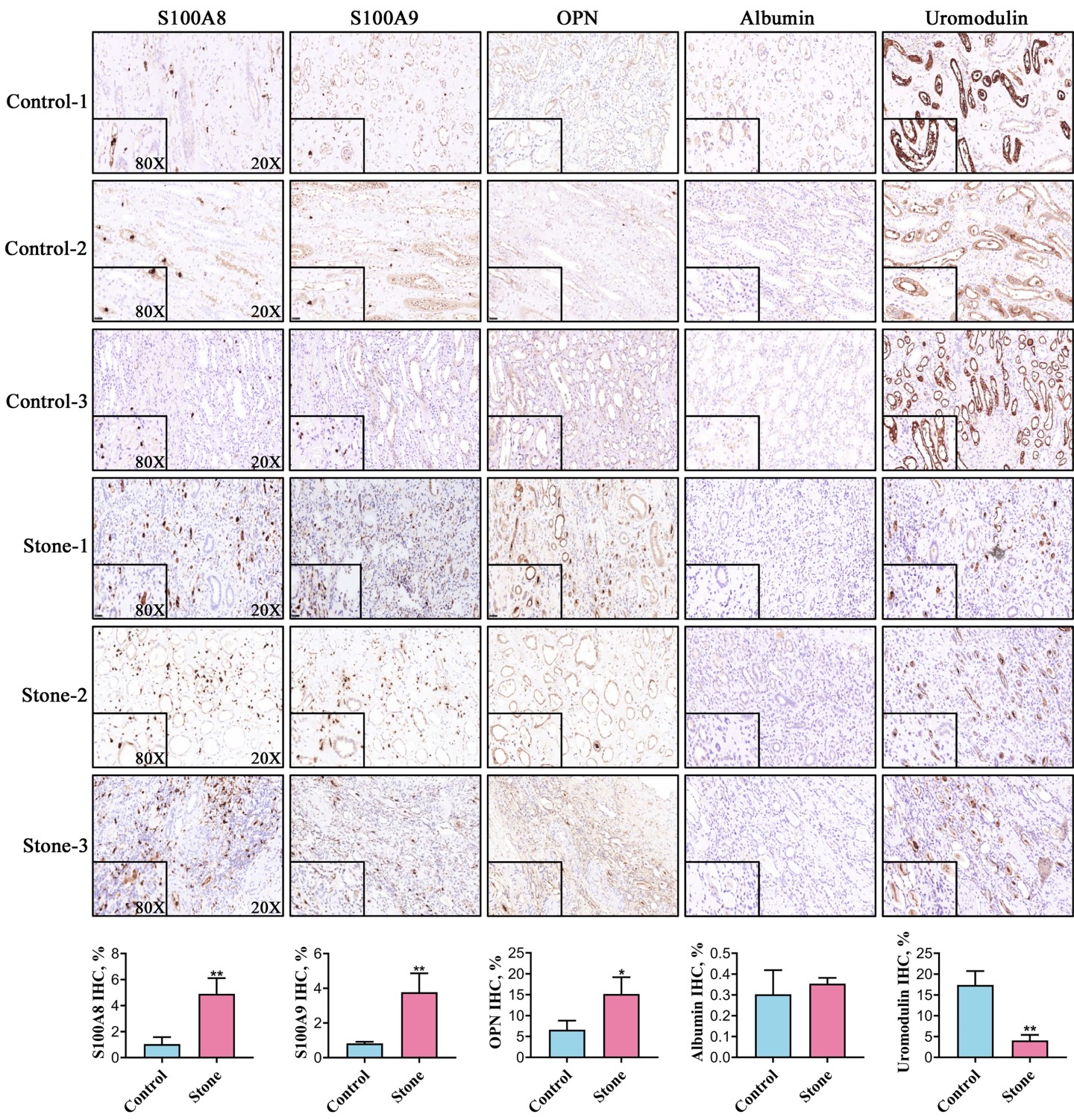

**Figure 5 Representative images for detection of the top five stone matrix proteins in renal tissue.** The renal expression of S100A8, S100A9, and osteopontin were increased, while uromodulin was decreased in kidney stone patients. Albumin was rarely expressed in kidney and there was no significant difference between healthy controls and kidney stone patients; $^*P < 0.05$ vs Control; $^{**}P < 0.01$ vs Control.

sialic acid and reducing its effectiveness in inhibiting stone formation (*Knörle et al., 1994*; *Viswanathan et al., 2011*). Increasing the expression of THP and restoring its function may be effective in the prevention of kidney stones.

Albumin is the fourth most common stone matrix protein and it is rarely expressed in the kidney. The albumin in the stone matrix was thought to originate from plasma proteins filtered through the glomeruli. Albumin is reported to be a powerful nucleator of COD and the polymer form is more active than the monomer form (*Cerini et al., 1999*). *Cerini et al. (1999)* thought that promotion of COD crystallization by albumin of crystallization mayight be a protective factor for urine stability, because the with rapid nucleation of small crystals caused the, the saturation levels to fall, preventing COM formation and aggregation with subsequent stone formation be prevented (*Cerini et al., 1999*).

OPN, the fifth most common stone matrix protein, is expressed in the distal renal tubules and thick ascending limbs of Henle's loop (*Xie et al., 2001*). The expression of OPN has been shown to increase in the kidneys of stone patients and experimental models, which was validated by our findings (*Kleinman, Wesson & Hughes, 2004*). The role of OPN in stone formation remains controversial and may depend on its phosphorylation level (*Wang et al., 2008*). *Wesson et al. (2003)* reported that OPN knockout mice developed more CaOx crystals in their kidney than wild type mice in the hyperoxaluria model. Their previous study reported that OPN favors the formation of calcium oxalate dihydrate (COD) over calcium oxalate monohydrate (COM). COD is less adherent to renal epithelial cells than COM, which may partly explain the antilithiatic effect of OPN (*Wesson et al., 1998*). However, other studies have shown that OPN may increase the risk of stone formation by promoting crystal adherence to the renal epithelium (*Yamate et al., 1996*; *Yamate et al., 1999*).

## CONCLUSIONS

We identified some common inflammation-related proteins that play a role in stone formation based on proteomic data, to help determine the pathogenesis of human urolithiasis. However, the mere detection of a protein does not explain how it participates in stone formation. Future studies are needed to identify the role of these proteins in stone formation and are expected to provide new diagnostic biomarkers and therapeutic targets for urolithiasis.

## LIST OF ABBREVIATIONS

**COM**      Calcium oxalate monohydrate
**COD**      Calcium oxalate dihydrate
**CaCO3**    Calcium carbonate
**CaOx**     Calcium oxalate
**CaP**       Calcium phosphate
**FDR**       False discovery rate
**GO**        Gene ontology
**HA**        Hydroxyapatite
**KEGG**    Kyoto Encyclopedia of Genes and Genome

| | |
|---|---|
| **LC-MS/MS** | Liquid chromatography-tandem mass spectrometry |
| **MAP** | Magnesium ammonium phosphate |
| **OPN** | Osteopontin |
| **THP** | Tamm-Horsfall protein |
| **UA** | Uric acid |

### Funding

This work was supported by the National Natural Science Foundation of China (81974092). The funders had no role in study design, data collection and analysis, decision to publish, or preparation of the manuscript.

### Grant Disclosures

The following grant information was disclosed by the authors:
National Natural Science Foundation of China: 81974092.

### Competing Interests

The authors declare that they have no competing interests.

### Author Contributions

- Yuanyuan Yang conceived and designed the experiments, performed the experiments, analyzed the data, prepared figures and/or tables, authored or reviewed drafts of the paper, and approved the final draft.
- Senyuan Hong performed the experiments, authored or reviewed drafts of the paper, and approved the final draft.
- Cong Li performed the experiments, authored or reviewed drafts of the paper, and approved the final draft.
- Jiaqiao Zhang performed the experiments, authored or reviewed drafts of the paper, and approved the final draft.
- Henglong Hu performed the experiments, authored or reviewed drafts of the paper, and approved the final draft.
- Xiaolong Chen analyzed the data, authored or reviewed drafts of the paper, and approved the final draft.
- Kehua Jiang analyzed the data, authored or reviewed drafts of the paper, and approved the final draft.
- Fa Sun analyzed the data, authored or reviewed drafts of the paper, and approved the final draft.
- Qing Wang conceived and designed the experiments, performed the experiments, analyzed the data, prepared figures and/or tables, authored or reviewed drafts of the paper, and approved the final draft.
- Shaogang Wang conceived and designed the experiments, authored or reviewed drafts of the paper, and approved the final draft.
## Human Ethics

The following information was supplied relating to ethical approvals (*i.e.*, approving body and any reference numbers):

Ethical Review Board of Tongji Hospital, Tongji Medical College, Huazhong University of Science and Technology have approved the collection and use of tissue samples (2019S1147).

## Data Availability

The detailed proteomic results of each study focusing on proteomics of kidney stones and the demographic characteristic of included patients and a subgroup analysis of the studies and focused only on CaOx stones are available in the Supplementary Files.

## Supplemental Information

Supplemental information for this article can be found online at http://dx.doi.org/10.7717/peerj.11872#supplemental-information.

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
