# Peer review of "Proteomic analysis reveals some common proteins in the kidney stone matrix"

_PeerJ, doi:10.7717/peerj.11872_

## Round 0.1 · original submission · Major Revisions

Please address the concerns of all reviewers and revise the manuscript accordingly.

Reviewer 1 ·

Basic reporting

The study by Yang et al., conducted a meta-analysis of previously published proteomic studies of urinary stones and crystals to identify the most common stone matrix proteins. They classified these common proteins according to their predicted functions and pathways/biological processes, and, finally, examined the over-expression of three of these proteins in the renal tissues of kidney stone patients using immunohistochemistry as a complementary technique to proteomics.

Generally, the manuscript is well-written, and the authors used clear, intelligible, and professional English for the majority of the manuscript. There have been, however, a few places where the authors need to express themselves more clearly. I have listed these cases for the authors under the “Comments for the author” section.

The presented study is short in its format, which is acceptable given the goals presented by the authors in their abstract and introduction of summarizing the literature and presenting common biomolecules to enable future nephrolithiasis research. The authors provided a sufficient background for their work and described how their work fit with the previous knowledge in the field. Overall, the study represents a cohesive body of work and is self-contained, with the presented results directly linked to the study goals described by the authors in their introduction.

The manuscript also follows the “Standard Sections” structure of the journal to a large extent except for the “Conclusions” section which was forced into the main text with a single sentence (please refer to my comments under the “Validity of the Findings” section for more details). The figures are of sufficient resolution, labelling, and description (except for the cases that I have listed for the authors under the “Comments for the author” section).

I commend that the authors shared the raw data in the main text tables and in the supplementary tables; the data were clearly described in the manuscript.

The authors have provided a “Statement of Ethics” in their manuscript and placed this statement under the “Materials and Methods” section as well per the journal’s instructions. The experiments in this study seemed necessary (to provide complementary data for the proteomic inferences) and ethical (the tissues were only obtained from patients undergoing nephrectomy) and the authors removed patient-identifying information from the text and supplementary/raw data. However, the “Informed consent form” and the “The approval document” are provided in Chinese. Please provide translated versions of these documents in English.

Experimental design

The present study represents an original primary research and fit under the “Biological Sciences” and “Medical Sciences” scopes of the journal. The authors explicitly stated their research questions, how their study contributes to filling the gap in previous knowledge of the field, and hence how their results are meaningful and relevant to the field.

The authors sought to follow high technical standards with regard to their immunohistochemistry experiment, controlling for baseline expression of their target proteins by including renal paracancer tissues in their protocol. The authors however didn’t justify the exclusion of uromodulin and albumin from the same experiment. Even if the patterns for these proteins were lower expression relative to the paracancer tissues or no expression at all, these results are useful for the comprehensiveness of the study (and at least serve as additional controls for the experiment). If conducting these additional experiments is not possible, the authors should mention this as a caveat of the study.

The study is generally meeting the ethical standards and the screening of >3,000 publications for the authors’ meta-analysis represents a rigorous investigation.

The authors need to describe their methods to sufficient detail and provide relevant information for other investigators to reproduce their results. For example, the authors need to provide more details on how they did the enrichment analyses and how they calculated the false discovery rate, indicating the specific software functions used for the different steps. They also need to clarify the criteria of exclusion for the 3,115 records displayed in Figure 1. Finally, the authors need to provide references for all the entries in Table 2 (as an additional column in the table).

Validity of the findings

As I have discussed under the “Basic reporting” section, the conclusions of this study were partially discussed by the authors at the end of the “Discussion” section (Lines 280-285), while the “Conclusions” section itself was forced into the main text with a single sentence. While the authors explicitly state their conclusions and link them to their original research questions, one of these conclusions is not really addressed by this study (beyond referencing literature). To elaborate, the authors didn’t do any experiments that address that there is a “common process” (Line 280) during stone formation. If the authors were referring to “inflammation and immune response” (Line 288) as the “common process”, then how would they fit the role of albumin (as a nucleator of calcium oxalate dihydrate) in this “common process”? If the authors chose to provide this “common process” as a speculation in the conclusion section, they should explicitly state so.

The authors provided all the underlying data, their experiment was controlled (although other controls could have been provided as discussed under the “Experimental design” section), and their bioinformatic analyses seemed thorough. However, in the absence of more details on the methodology behind the enrichment analyses (as discussed under the “Experimental design” section), it’s hard to assess the soundness of the statistics applied.

Additional comments

Line 136: What is the meaning of “protein files”?
Line 185: The citation linked to “Thurgood et al.” is missing from the text. This happened in several places throughout the manuscript. Please make sure to provide the citation numbers in the text.
Line 208: What is meant by “calcium iron”?
Line 235: “It was surprising that only four of the 20 proteins showed medium to high expression in kidney”. Which table/figure is referred to here?
Line 258: The sentence should start with “In virto”
Lines 258-266: Please use the past tense when reporting the results from previous studies (in these lines as well as other places throughout the manuscript). Also, please revise the text for the cohesiveness between the sentences.
Lines 280-282: The expressions in these lines are very vague; please rewrite more clearly
Figure 2: Panel A should read “Detection frequency”
Figure 3: Please spell out “BP” and “MF”. Also, please describe in the legend the “Rich Factor” and the “FDR”, indicating the FDR cut-off used on the figure.
Figure 4: the x-axis should read “-log(Pvalue)”. Why the FDR was not used here? What is the cut-off?
Figure 5: Please spell out “Ctrl”

Reviewer 2 ·

Basic reporting

No commenrts

Experimental design

No comments

Validity of the findings

No comments

Additional comments

The manuscript entitled "Proteomic analysis of stone matrix: A window to the pathogenesis of urolithiasis' is very well defined, written, and presented. The authors used bioinformatic tools and identify 20 top proteins to explore their biological function and performed Immunohistochemistry of the three main matrix proteins.

The results are very nicely presented, and I don't have much to add however, the author may like to reduce the introduction section of the abstract and may increase the results section.


Thank you

Reviewer 3 ·

Basic reporting

Introduction: The introduction suffers from some grammatical errors. It should read “Kidney stone disease is a common…” and “10 to 12% of people”. I will not comment further because this text will likely change with the restructuring of this paper.
Title: The title of the paper appears to be quite similar to reference 26. The title does not describe the work done by the authors, but work performed in other labs. A better title may be “Calgranulin and osteopontin in calcium oxalate urolithiasis: Roles for inflammation and immune response.”
Overall review: This paper begins and reads as a review article with the inclusion of primary histologic data on osteopontin and S100 proteins in kidney biopsies of only calcium oxalate stone patients. The paper would be better restructured to start with the author’s primary immunohistochemistry findings in the renal biopsies from calcium oxalate patients.
The authors probed for calgranulins and osteopontin, likely due to prior interest in these critical proteins in their research lab (References 28, 36). These proteins have been studied for quite some time for their role in kidney stone disease and inflammatory diseases throughout the body (calcific-arthritic conditions, cardiac calcifications). The primary findings of increased expression of these proteins in the kidney tissue of calcium oxalate stone patients have been largely overshadowed by the extensive literary review of kidney stone matrix proteins.
The articles reviewed should only include those that look at the stone matrix from calcium oxalate kidney stones because this is consistent with the immunohistochemistry study population. Thus, there would be 12 studies of interest (References 8, 9, 10, 15, 16, 19, 20, 21, 22, 24, 25, and 26), choosing to study only the calcium oxalate data only when a paper reports more than one stone type. Comparing all types of stone matrix into one summation obscures the differences that are present in varying stone disease states. It actually impedes urolithiasis research and the goal of understanding disease pathology by assuming that all stones are formed due to the same type of inflammation and immune responses. The authors might also consider the removal of the protein-crystal binding studies (Ref 13, 14) from the comparison since these are derived from non-stone forming participant urine samples.
Again, the authors hide their primary findings behind the review of proteomic papers. This is doing a disservice to the actual findings in the paper. Little data has been generated on kidney stone patient biopsy samples due to the limited availability of such samples. Is there any other quantitative data that may be relevant to the patients studied in this paper (Demographics, comorbid health histories such as hypertension, diabetes). Additionally, is there any way that the immunohistochemistry work could be quantified. Regions of interest (ROI) quantification is commonly available in software packages. This would strengthen the visual observation of increased expression of osteopontin and S100 proteins.
Line 204: Biologic GO and KEGG annotation is a good set of analysis. It should be repeated with the proteins derived from studies relevant to your calcium oxalate stone forming study population (Likely the 12 studies highlighted above). This new comparison may or may not change the proteins listed in Table 2.
Line 274: Osteopontin’s role in stone disease is confounded by the level of phosphorylation of the protein which may be decreased in kidney stone formers. Tamm Horsfall protein similarly will have a different behavior depending on the level of glycosylation (sialic acid) on the protein (Viswanathan, et.al.).
Lines 276-277: The discussion of albumin as a monomer or a polymer is a confusing way to describe a moderate sized protein.

Experimental design

Seems OK.

Validity of the findings

Suggestions have been made above.

Additional comments

This manuscript needs a major rewrite and restructuring before it should be published. Suggestions have been made above. Grammar and incorrect or missing words need to be cleaned up.

---

## Round 0.2 · accepted · Accept

All critiques were successfully addressed and the manuscript was amended accordingly. Therefore, the revised version is acceptable now.

Reviewer 1 ·

Basic reporting

The authors have adequately addressed my concerns, provided additional experimental results and supplementary data, and made appropriate and thorough revisions in the resubmitted manuscript.

Experimental design

No further comments.

Validity of the findings

No further comments.

Additional comments

Thank you for providing this resource and for thoroughly addressing my comments.

Reviewer 2 ·

Basic reporting

No further comment

Experimental design

No further comment

Validity of the findings

No further comment

Additional comments

The manuscript is quite improved now and I don't have any other comments